# A New Look at the Role of Radiation-Related Epigenetic Mechanisms in Diagnosis and Anticancer Therapies

**DOI:** 10.3390/cells14231885

**Published:** 2025-11-27

**Authors:** Adam Jan Olichwier, Magdalena Bruzgo-Grzybko, Izabela Suwda Kalita, Natalia Bielicka, Ewa Chabielska, Anna Gromotowicz-Poplawska

**Affiliations:** 1Radiopharmacy Centre, Medical University of Bialystok, 15-569 Bialystok, Poland; 2Department of Biopharmacy and Radiopharmacy, Medical University of Bialystok, 15-222 Bialystok, Poland

**Keywords:** epigenetics, radiation, cancer, RNA and DNA methylation, histone modifications, anticancer

## Abstract

Epigenetics encompasses heritable but reversible modifications of gene expression that occur without changes in the DNA sequence and involve mechanisms such as DNA and RNA methylation and histone modifications. These mechanisms modulate chromatin architecture, genome stability, and cellular responses to environmental stressors, and their dysregulation contributes to oncogenesis and cancer progression. In parallel, radiotherapy remains a cornerstone of cancer treatment; furthermore, ionizing radiation induces epigenetic modifications alongside direct DNA double-strand breaks and oxidative damage. Radiation-induced epigenetic changes, including global or locus-specific DNA methylation shifts (e.g., genes promoter CpG islets), histone acetylation and methylation imbalances, are increasingly recognized as key contributors to molecular radioresistance. These adaptive responses may enhance tumor cell survival, affect therapeutic efficacy, and promote metastasis. Understanding the interplay between radiation exposure and epigenetic remodeling opens new perspectives for precision oncology and diagnostics. Epigenetic biomarkers hold potential for predicting treatment response and prognosis, while epigenetic modifiers may sensitize tumors to radiation. This review summarizes current evidence on radiation-induced epigenetic mechanisms and evaluates their diagnostic, prognostic, and therapeutic implications in cancer management.

## 1. Introduction

Epigenetics is a mechanism of cellular regulation that controls gene expression without altering the DNA sequence [1,2,3,4,5]. Through mechanisms such as DNA and RNA methylation, histone modifications, or non-coding RNA activity, epigenetic regulation controls chromatin structure, genome stability, gene expression, and cell response to various signals [1,6,7,8]. Importantly, epigenetic changes have been shown to play a role in the development and progression of many diseases, e.g., cancers [9]. The dynamic and reversible nature of epigenetic modifications has made them attractive targets for both diagnostic and therapeutic approaches.

At the same time, radiopharmacy focuses on the development and use of new radioactive drugs—radiopharmaceuticals, for both diagnosis and treatment [10]. Molecular imaging based on radiopharmaceuticals can help visualize biological processes at the cellular and molecular levels, thus enabling the early detection of diseases. It can also be used to target specific cells or tissues, potentially delivering radiation to destroy cancer cells or modulate gene expression [11]. On the other hand, cancer therapy can also involve the use of external sources of radiation, such as in external beam radiation therapy using high-energy photons, electrons, or protons [12]. Exposure to radiation—whether from radiopharmaceuticals or from external sources used in radiotherapy—can induce epigenetic changes within cells. In cells after radiation exposure, one of the most common events is the occurrence of epigenetic modifications, which are involved in the regulation and function of key genes and proteins in various radiation response pathways, and may also promote mechanisms of radiation resistance [12]. These radiation-induced effects often manifest as alterations in DNA methylation, histone modifications, and changes in non-coding RNA expression, all of which play critical roles in the regulation of gene expression and cellular response to radiation. Radiation directly damages DNA through the induction of double-strand breaks (DSBs) and indirectly through the formation of reactive oxygen species (ROS) [12,13]. Molecular radioresistance comprises genetic and epigenetic characteristics that are either inherent to cancer cells or acquired after exposure to radiation. This approach aims to deliver focal radiation to the tumor and surrounding at-risk tissue compartments to inhibit tumor growth, proliferation, and metastasis [12].

Despite growing evidence that radiation induces profound epigenetic alterations affecting both therapeutic response and normal tissue toxicity, the current literature largely treats radiobiology, epigenetics, and radiopharmacy as separate domains, leaving a fragmented understanding of how these mechanisms intersect in clinical practice. Existing reviews tend to focus either on molecular epigenetic changes or on radiopharmaceutical advancements, but rarely integrate these perspectives to explain how radiation-related epigenetic remodeling can be harnessed for both improved diagnosis and targeted anticancer therapy. This review synthesizes both fields to outline how radiation-triggered DNA/RNA methylation changes, histone remodeling, and enzyme dysregulation can serve as actionable biomarkers or therapeutic targets within nuclear medicine. It uniquely emphasizes the emerging potential of radiotheranostics targeting epigenetic enzymes and demonstrates how understanding radiation-epigenome interactions can guide the development of new diagnostic tracers and epigenetically informed radiosensitizing strategies. This interdisciplinary framing—bridging epigenetic biology with radiopharmaceutical innovation—constitutes the main conceptual advance of the review. This work addresses that gap by providing a unified framework that links radiation-induced epigenetic mechanisms with their implications for radiotheranostics, radiosensitization strategies, and radiation-associated toxicities. By combining a radiopharmacy perspective with emerging epigenetic insights, we highlight new opportunities for precision imaging, personalized treatment planning, and development of epigenetically informed radiotherapeutic approaches.

By understanding how radiotherapy and radiopharmaceuticals affect the epigenome, researchers aim to develop new therapeutic strategies for diseases. This could involve targeting specific epigenetic pathways to reverse detrimental changes or promote beneficial ones. In this review, we will try to summarize radiation-related epigenetic changes, such as DNA methylation and histone modifications (namely, acetylation and methylation), and how these changes can be used as diagnostic or prognostic tools in patients with cancers.

## 2. DNA/RNA Modifications

Epigenetic modifications of RNA and DNA have been shown to play an important role in tumorigenesis and diagnosis [14]. The main DNA-related epigenetic modifications include DNA methylation and histone modifications such as methylation, acetylation, phosphorylation, and ubiquitination [14,15]. RNA modifications include N6-methyladenosine (m6A), pseudouridine (Ψ), 5-methylcytosine (m5C), 2′-O-methylation or N7-methylguanosine (m7G) [16,17,18]. RNA and DNA modifications affect genome stability, gene expression, protein synthesis, cellular processes, immune function and regulation of repair processes [19,20]. Importantly, epigenetic modifications of RNA and DNA can be modified by radiation [21,22].

Indirect actions of radiation involve the formation of ROS, which can affect gene expression. Ionizing radiation (IR) induces DSBs both through direct high-energy damage to DNA and through free radicals generated within cells. ROS can also affect DNA by, e.g., oxidizing nucleoside bases, leading to the formation of 8-oxo guanine [23]. They can alter nucleotide sequences, dysregulate base excision repair (BER) control and repair action [24], and cause mitochondrial DNA lesions, strand breaks, and degradation of mitochondrial DNA [13,23,25]. ROS production has been implicated in radiotherapy due to its effect on downstream cell survival [26], death signaling [23,27], primary prevention [28,29], and enhancement of therapeutic responses [30,31]. Importantly, ROS can directly induce DNA and histone modifications, affect the activity of epigenetic enzymes, and influence the expression of non-coding RNAs, leading to global changes in DNA methylation [7].

In the following sections of this review, we will focus on RNA and DNA methylation in the context of radiation exposure and their roles in cancer diagnosis and therapies.

### 2.1. RNA Methylation

RNA methylation is an epigenetic modification that can regulate gene expression without altering the DNA sequence. Methyl groups can be added or removed by specific enzymes: RNA methyltransferases (RNMTs), which use S-adenosylmethionine (SAM) as a methyl-group donor to transfer the methyl group to RNA sequences [1,2], and RNA demethylases, which reverse these modifications and restore RNA to its original state [2,32]. Methylation modifications in RNA sequences can also be recognized by specific proteins, which affect various cellular mechanisms such as gene expression and cell signaling [2,33]. RNA methylation affects several biological processes, including RNA splicing, stability, translocation, cell growth, immunity, and aging [2,33,34,35]. These modifications are primarily found in messenger RNA (mRNA) and long non-coding RNA, where they play essential roles in various cellular functions and disease mechanisms, but they are also widespread across other RNA types, including transfer RNA (tRNA), ribosomal RNA (rRNA), and small nuclear RNA (snRNA) [18,36].

#### 2.1.1. RNMTs (METTL3 and METTL14)

The most abundant and well-studied type of RNA modification is m6A, which plays a significant role in the regulation of gene expression at the post-transcriptional level [37,38]. m6A is involved in various processes of RNA metabolism, such as stability, splicing, transcription, translation, and degradation [38]. It is a reversible epigenetic modification, a type of methylation that occurs in non-coding RNAs in tumors [38,39] and is implicated in cancer progression [40]. The best known RNMTs include methyltransferase-like 3 (METTL3) and METTL14, which are core components of the methyltransferase complex responsible for adding a methyl group to RNA sequence to form m6A [41,42]. Catalytically active subunit responsible for methylation is METTL3, while METTL14 is a structural component that stabilizes METTL3 and helps in recognizing and binding the RNA substrate [42]. METTL3 is involved in tumor progression, invasion, growth, and drug resistance in various cancers [43]. METTL3 was shown to be involved in UV-induced DNA damage response in U2OS (osteosarcoma) and HeLa (cervical cancer) cells, where elevated METTL3 recruitment was observed [44]. Similarly, carbon-ion radiotherapy elevates METTL3 levels in non-small-cell lung cancer (NSCLC), and METTL3 knockdown impairs the proliferation, migration, and invasion of NSCLC cells, both in vivo and in vitro, by inhibiting H2A histone family member X (H2AX) mRNA degradation, which leads to enhanced DNA damage repair and increased cell survival [45]. Similarly, METTL14 was shown to enhance the expression of miR-99a-5p, which inhibits the persistence of cancer stem-like cells (CSCs) and reduces the radioresistance of esophageal squamous cell carcinoma (ESCC). Downregulation of miR-99a-5p was associated with an unfavorable prognosis in ESCC patients [46]. Furthermore, METTL14 level was shown to be involved in further facilitating the persistence of CSCs in ESCC by suppressing *tribbles homolog 2* (*TRIB2*) expression [46]. TRIB2 is a scaffold protein involved in cell proliferation, survival, differentiation, and apoptosis through the regulation of MAPK and AKT signaling pathways [47]. METTL14 depletion was also shown not only to enhance cell resistance to highly active agents used in the treatment of advanced breast cancer (i.e., paclitaxel and doxorubicin) but also to confer radioprotective effects on ALDH+ breast cancer stem cells [48]. These findings highlight the role of METTLs in radioresistance and suggest potential therapeutic targets for cancer treatment.

#### 2.1.2. RNA Demethylases (ALKBH5 and FTO)

On the other hand, RNA demethylases include, for example, alkB homolog 5 (ALKBH5) [49] and fat mass and obesity-associated protein (FTO) [50], which are responsible for removing m6A modifications from mRNA and non-coding RNAs. In patient-derived glioblastoma stem cells (GBMSCs), where *ALKBH5* expression was downregulated, decreased GBMSC survival after irradiation was observed due to defects in DNA damage repair [51]. Simultaneously, a decrease in the expression of genes involved in homologous recombination, such as *checkpoint kinase 1* (*CHK1*)—a crucial serine/threonine kinase in the DNA damage response [52]—and DNA repair protein *RAD51 homolog 1* (*RAD51*), essential for repairing double-strand breaks in DNA [53], was also observed [51]. In cervical squamous cell carcinoma (CSCC), the m6A demethylase FTO promotes resistance to chemoradiotherapy by upregulating β-catenin through mRNA demethylation, which in turn enhances excision repair cross-complementation group 1 (ERCC1) activity [54]. Upregulation of FTO in radioresistant nasopharyngeal carcinoma (NPC) tissues and cells, relative to their parental radiosensitive parts, was also observed [55]. FTO by m6A modification of the OTU deubiquitinase, ubiquitin aldehyde binding 1 (OTUB1) transcripts and promotes the expression of OTUB1—a protein involved in regulating protein stability, cell proliferation, immune responses, energy metabolism, and DNA damage [56]. This mechanism inhibits radiation-induced ferroptosis in NPC cells [55]. In both in vitro and patient-derived xenograft models, treatment with the FTO inhibitor FB23-2 and the ferroptosis activator erastin enhanced the sensitivity of NPC cells to radiotherapy [55]. In HPV head and neck squamous cell carcinoma (HNSCC) models in humans and mice, genetic and pharmacological inhibition of FTO enhanced the efficiency of radiotherapy [57]. This effect was associated with increased DNA damage, reduced efficiency of homologous directed repair, and decreased formation of RAD51 foci. Importantly, FTO inhibitors do not exacerbate radiation-induced oral mucositis, a common toxic effect of HNSCC treatment [57]. Therefore, RNA demethylases such as ALKBH5 and FTO clearly play important roles in cancer therapy and radiosensitivity, and they represent promising targets for the development of more effective therapeutic strategies to improve treatment outcomes for patients. In Table 1, we summarize all RNA modifications mentioned in the main text.

### 2.2. DNA Methylation

DNA methylation is a dynamic and reversible modification of DNA. A methyl group is added to the fifth carbon position of the cytosine ring in DNA (5-methylcytosine) in a process regulated by DNA methyltransferases (DNMTs) [3,4,5]. This process can silence genes, prevent their activation, and play an important role in cell differentiation, development, and the maintenance of genome stability [2]. Importantly, DNA methylation can modify gene expression without changing the DNA sequence and can be passed from one cell to another, mainly during the cell cycle, from the parental to the daughter cell [58,59]. DNA methylation, as a dynamic process, can be influenced by diet, environmental exposures, age, or other signals from outside or inside the cell [60,61].

#### 2.2.1. ROS and DNA Methylation

Radiation, as one of the main forms of cancer treatment, triggers DNA damage through different mechanisms, like ionization or ROS production. These changes lead to the destruction of DNA structure, DSBs, transcription complex, and polymerase dysfunction, all of which consequently result in tumor cell death [62,63,64]. ROS have been shown to downregulate *AT-rich interacting domain 1 A* (*ARID1A*) expression, a key protein kinase that plays a crucial role in maintaining genomic stability by responding to DNA replication stress and DNA damage. ARID1A is an AT-rich interacting domain 1A of brahma-related gene 1 (BRG-1)-associated factor complex (BAF), chromatin remodeler commonly mutated in cancer. What is important, ROS were found to decrease *ARID1A* expression through promoter methylation in ovarian cancers [65,66]. This provides clear evidence of the connection between ROS and DNA methylation.

#### 2.2.2. DNMT1

Another example of radiation affecting DNA methylation was observed in the murine thymus, where low-dose X-ray irradiation increased global DNA hypomethylation, increased accumulation of DNA damage, and reduced expression of DNMTs [67]. Changes in DNA methylation in response to radiation have also been reported in malignant cancers, and these were associated with cellular sensitivity to radiotherapy and DNMT levels [68]. In the more malignant MDA-MB-231 human breast cancer cells (compared with MCF-7), loss of DNA methylation and altered expression of *DNMT1*, as well as *methyl-binding proteins methyl-CpG binding protein 2* (*MeCP2*) *and methyl-CpG binding domain protein 2* (*MBD2*), were observed relative to the non-tumorigenic MCF-10-2A epithelial breast cells [69]. These findings indicate that epigenetic differences among breast cancer tumor cell lines may contribute to and serve as an indicator of the development of a more aggressive tumor phenotype during tumor progression.

#### 2.2.3. DNMT3A/B

In rhabdomyosarcoma (RMS), the most common soft tissue sarcoma in childhood, primary tumor biopsies have shown overexpression of *DNMT3A* and *DNMT3B* [70]. In embryonal RMS cells, knockdown of DNMT3A and DNMT3B followed by IR exposure led to increased radiosensitivity. Silencing of *DNMT3A* activates the senescence program by upregulating p16 and p21 [71], cyclin-dependent kinase inhibitors involved in the induction and maintenance of cellular senescence [72], whereas *DNMT3B* silencing induces significant DNA damage and impairs the DNA repair mechanisms [71]. These findings suggest that *DNMT3A* and *DNMT3B* expression can influence the efficacy of radiotherapy and that their inhibition may represent a promising radiosensitizing strategy, particularly for RMS patients. In NPC, elevated *DNMT3B* expression was noticed after exposure to irradiation, which was associated with the development of radioresistance. On the other hand, *DNMT3B* silencing in response to radiation, mediated by DNA demethylation, enhanced the activation of cell cycle regulators p53 and p21, led to G1 phase of cell cycle arrest, and promoted apoptosis [73]. These findings open new possibilities in developing novel therapies using epigenetic modulations and radiation in NPC. In prostate cancer cells, PC3 cells, *DNMT3B* expression was shown to be induced by X-ray radiation. Downregulation of *DNMT3B* by miR-145 was achieved through direct targeting, and DNMT3B knockdown increased expression of miR-145 by CpG island promoter hypomethylation, consequently sensitizing prostate cancer cells to radiation [74]. These results further support the possibility of a combination of epigenetic regulation and radiation as a combined therapeutic strategy in prostate cancer. Therefore, we can conclude that DNMT3A primarily regulates the senescence program in RMS, with its silencing increasing *p16* and *p21* expression and promoting cellular senescence, while DNMT3B is more directly linked to DNA damage responses and repair, and its silencing leads to impaired DNA repair and enhanced radiosensitivity across multiple cancer types.

#### 2.2.4. DNA and CpG Island Methylation

Interestingly, epigenetic mechanisms have also been shown to play a crucial role in the arginosuccinate synthetase 1 (ASS1)-negative glioblastoma (GBM) tumor model. Treatment of (ASS1)-negative GBM with ADI-PEG20, an inhibitor of arginase 1 (Arg1), was shown to be effective, whereas ASS1-positive GBM cells were unaffected by ADI-PEG20. This differential response was associated with the absence of CpG island methylation, and only the combination of radiotherapy and ADI-PEG20 achieved complete tumor elimination by promoting macrophages/microglia recruitment and M1 repolarization [75]. In another study, biopsies from patients with inoperable breast cancer before and after radiotherapy (patients receiving 10–24 Gray) were analyzed. Eighty-two differentially methylated genes were observed in irradiated samples (compared to non-irradiated), predominantly in genes involved in immune system pathways. Among the genes significantly associated with clinical response and altered methylation were: *H2A histone family, member Y* (*H2AFY*), involved in the epigenetic response [76]; *cathepsin A* (*CTSA*), responsible for degradation intra- and extracellular substrates [77]; *leukotriene C4 synthase* (*LTC4S*), participating in the synthesis of proinflammatory lipid mediator [78]; *interleukin 5 receptor subunit alpha* (*IL5RA*), involved in immune response regulation [79]; and *RB transcriptional corepressor 1* (*RB1*), a key cell cycle/tumor suppressor [80]. All of these genes are associated with tumor progression. Interestingly, the degree of methylation correlated with the radiation dose and was enriched for genes involved in immune response, proliferation and apoptosis pathways [81]. The data mentioned above show that changes in DNA methylation in tumors following radiotherapy correlate with methylation levels, may be dose-dependent, and can be associated with treatment response in breast cancer.

#### 2.2.5. DNMTis

Another strategy used in radiation-related therapies involves DNMT inhibitors (DNMTis). Treatment with 3 DNMTis (psammaplin A, 5-aza-2′-deoxycytidine, and zebularine) in combination with radiation exposure was shown to induce radiosensitivity in both A549 (lung cancer) and U373MG (glioblastoma) cell lines, was observed [6]. Furthermore, psammaplin A increased the sub-G1 fraction of A549 cells, and prolonged H2AX expression, a marker of radiation-induced DSBs [82]. This was observed in the cells treated with DNMTis prior to radiation compared with those exposed to radiation alone, suggesting an association with the inhibition of DNA repair [6]. Gamma-phosphorylated H2AX (γH2AX) recruits multiple components of the DNA damage response to sites of DNA DSBs to initiate repair. Therefore, elevated ROS, which was previously observed to affect γH2AX phosphorylation, can blunt treatment response to radiation as well as worsen outcomes for some cancers [13]. In another study, a newly synthesized phthalimido-alkanamide derivative, MA17, which is a DNMTi, was used in glioblastoma cell lines (U87MG, U373MG, U138MG, and T98G) and normal human astrocytes (NHA). MA17 was shown to radiosensitize all glioblastoma cell lines without affecting normal astrocytes. Furthermore, DNMT activity and expression of the *FA complementation group A* (*FANCA*; involved in DNA repair and maintenance) genes were downregulated. Increased apoptosis, autophagy, and DSBs formation were also observed in MA17-treated cells compared with those subjected to radiotherapy alone [83]. These findings demonstrate that DNMTis exerts promising radiosensitizing effects and may represent effective adjuvant agents to enhance the efficacy of radiotherapy in cancer treatment.

#### 2.2.6. Radiation-Driven Epigenetic Changes

Radiation used in cancer therapy has been shown to be connected with the possibility of developing radiation-induced cardiovascular disease (RICVD) [84]. As a mechanism involved in this process, next to the endothelial inflammatory activation, premature endothelial senescence, increased ROS generation, mitochondrial dysfunction, and epigenetic mechanisms, particularly DNA methylation, are pointed out [84,85]. RICVD represents a significant problem in thoracic radiotherapy [84]. In heart-irradiated rats exposed to 27.6 Gy of radiation, DNA methylation alterations persisted up to 7 months after exposure, with differential expression of cardiac-relevant differentially methylated regions [84]. Higher expression of the *sarcolemma-associated protein* (*SLMAP*) gene, involved in cell cycle control, migration and signaling [86], was correlated with hypomethylation [84]. In contrast, expression of the *E2F transcription factor 6* (*E2F6*) gene, which regulates the cell cycle and is involved in chromatin modification and DNA methylation [87], was inversely correlated with a decreased global longitudinal strain [84] in these rats. Similarly, in radiotherapy-treated breast cancer patients, *E2F6* and *SLMAP* exhibited differential expression directly and six months after radiotherapy, respectively [84]. These findings suggest a direct association between radiation-induced DNA methylation changes and the pathophysiology of RICVD, which could serve as a basis for future diagnostic or therapeutic approaches. The wild-type isocitrate dehydrogenase (IDH) enzyme supports DNA demethylation by producing α-ketoglutarate (α-KG) [88]. However, mutations in *isocitrate dehydrogenase 1* and *2* (*IDH1/2*) are observed in approximately 80% of gliomas. These mutations lead to the production of oncometabolite d-2-hydroxyglutarate (d-2-HG), which results in aberrant DNA methylation. Simultaneously, IDH mutant gliomas are more sensitive to radiotherapy due to reduced levels of NAD and NADPH, which are required to counteract oxidative stress [89]. This was demonstrated in a phase III clinical study in patients with IDH mutations/non-codel tumors treated with radiotherapy and Temozolomide (TMZ). Patients with high-risk low-grade glioma showed longer progression-free survival (median PFS 55 months versus 36 months; *p* = 0.0043) after radiotherapy than those treated with TMZ alone [90]. These findings suggest that DNA demethylation can also play a crucial role in cancer-related treatment and radiation response.

DNA methylation is a dynamic but heritable epigenetic modification that can be reshaped by radiation and ROS, influencing gene expression, genome stability, and cellular radiosensitivity. Radiation alters the activity of DNMTs, modulates CpG island methylation, and drives gene-specific methylation changes that can either promote tumor progression or enhance radiosensitivity, depending on the cellular context. These radiation-induced methylation patterns not only affect cancer treatment outcomes through DNMT1, DNMT3A/B, and DNMTis-related mechanisms but also contribute to long-term normal-tissue effects such as radiation-induced cardiovascular disease. As a summary, we gathered together all the mentioned DNA modifications, related to anticancer treatment, mentioned in this paragraph in Table 2. We specified the mechanism of action, used cell line or model, and how epigenetic modification was modified in relation to radiation.

## 3. Histone Modifications

Chromatin is a complex of DNA and proteins (histones) that forms chromosomes within the nucleus of eukaryotic cells. Histones are small proteins that help pack long DNA strands into compact structures. Five types of histones—H1, H2A, H2B, H3, and H4—are distinguished; they are rich in basic amino acids (arginine and lysine), which promote binding to the negatively charged DNA [92]. Histone modifications are crucial not only for chromatin structure and remodeling but are also involved in DNA repair, tagging of DSBs, activation of repair regulators, cell cycle regulation, and apoptosis [7,93,94]. In preoperative chemoradiotherapy, epigenetic therapies—including DNA methylation and histone deacetylase inhibitors (HDACis)—sensitize tumors to radiopharmaceuticals, enhancing treatment synergy. Early clinical data show that the combined use of epigenetic markers and radiopharmaceuticals improves therapeutic efficacy, increases survival rates, reduces recurrence, minimizes off-target effects, enhances tumor selectivity, and facilitates tumor downstaging [95]. Elevated ROS levels have also been associated with histone protein degradation and decreased phosphorylation of H2AX, a key step in the DNA repair process. Moreover, radiation also induces histone modification, i.e., phosphorylation of histone H2AX, namely, phosphorylation at serine 139 (γ-H2AX), that is used as a measure of DSBs in early caused cellular response to IR and is crucial for repair of DSBs [7]. Moreover, ROS were shown to be able to activate enzymes involved in histone control and have a dual effect on DNMTs, which play a role in the expression of oncogenes and tumor suppressor genes [25,91]. Therefore, in the following section, we will focus on two main histone modifications, such as histone methylation and acetylation, in cancer, with particular attention to their roles in radiation-related treatments. All histone modifications discussed in the following paragraphs in the context of anticancer treatment and diagnostics are summarized in Table 3.

### 3.1. Histone Methylation

Histone methylation is an epigenetic process involving the addition of a methyl group to lysine or arginine amino acids, on the N-terminal “tails” of histone proteins [8,116]. This process is regulated by two classes of enzymes: histone methyltransferases (HMTs), which catalyze the addition of methyl groups, and histone demethylases (HDMs), which remove them [8,117]. More than one methyl group can be added on lysine residues, mono (me1), di-(me2) or tri-(me3) and me1 or me2 on arginine [118]. Histone methylation modulates DNA–histone attraction, chromatin structure, DNA access, and transcription [8,116]. Methyl groups on histones also act as docking sites for other proteins involved in cell cycle, function, and signaling [119]. Histone methylation can have both effects on gene expression, depending on the specific histone residue [120]; e.g., H3K9me3 is associated with the formation of “closed” chromatin and blocking expression [121], while H3K4me3 leads to gene expression and “open” chromatin [108]. This dynamic process is involved in multiple biological functions, including DNA repair [122], cell cycle control [123], aging [116,123], and stress response [116]. Importantly, histone methylation is involved in the regulation of gene expression in response to radiation. Radiation caused by DNA damage also affects chromatin structure and histone methylation and can impact response to DSBs. Changes in histone methylation can affect long-term repair processes and affect cells’ radiosensitivity [124,125]. Therefore, histone methylation represents a potential target for cancer therapy, improves radiotherapies, or contributes to tumor progression and adaptation.

#### 3.1.1. X-Ray Irradiation and Histone Methylation

Researchers have shown that even low-dose X-ray irradiation in an in vitro murine model decreased me3 of histone H4 in the thymus and reduced chromatin compactness [67]. In the MDA-MB-231 breast cancer cell line cells, observations included decreased 3me of lysine 20 of histone H4 and hyperacetylation of histone H4, compared to MCF-7 cells, as well as lower expression of *suppressor of variegation 4–20 homolog 2* (*Suv4-20h2*) and *histone methyltransferase*. These changes were associated with increased malignant properties of breast cancer cells [69]. The presented data demonstrate that significant epigenetic alterations linked to histone methylation occur in human breast cancer cells, while X-ray irradiation induces histone methylation modifications in murine models. These observations suggest that histone methylation can be used as a novel target for many different treatments.

#### 3.1.2. HDMs

The H3K4-specific histone lysine demethylase 5 (KDM5) has been shown to be overexpressed in different cancers, including breast, prostate, lung and bladder carcinomas [33,96,97]. KDM is involved in transcriptional regulation and DNA repair and, in cancer, exerts positive effects on cancer proliferation and chemoresistance [33,126,127]. In MCF-7 breast cancer cells overexpressing *KDM5B*, also known as *JARID1B*, inhibition of KDME activity increased the sensitivity of breast cancer cells to IR and enhanced radiation-induced damage by blocking the catalytic function of KDM5 enzymes [96]. Similarly, in ESCC patients presenting overexpression of *KDM5B*, its inhibition enhanced the H3K4me3 methylation of phosphatidylinositol 3-kinase catalytic subunit type 3 (PIK3C3) promoter and induced the expression of *PIK3C3*. In in vitro models (KYSE-150 and TE-10 cells), knockdown of *KDM5B* promoted apoptosis, cell cycle arrest, autophagy, and increased sensitivity to radiotherapy [97]. MiR-320a regulates a number of genes involved in various physiological processes, including tumor suppression [128]. In non-small cell lung cancer (NSCLC) clinical samples, suppressed expression of *KDM5B* and enhanced radioresistance of NSCLC through the downregulation of *phosphatase and TENsin homolog* (*PTEN*) expression, a crucial tumor suppressor that inhibits cell growth, proliferation, and survival [129], was reported to be caused by KDM5B [98]. Furthermore, inhibition of miR-320a in radioresistance of NSCLC was also reproduced by in vivo assay [98]. Therefore, the role of KDM5, an epigenetic enzyme, seems to be crucial for radioresistance and cancer treatment, and future directions targeting this histone methylation-related enzyme can be a promising new branch in pharmacy.

### 3.2. Histone Acetylation

In histone acetylation, an acetyl group is added to lysine residues on histone proteins, which neutralizes their positive charge, loosens the DNA–histone interaction, and consequently causes the chromatin to loosen, relaxed structure, calling “open” for interaction with transcription factors, which leads to increased gene expression [130]. Histone acetylation is regulated by two main classes of enzymes: histone deacetylases (HDACs) and histone acetyltransferases (HATs) [130,131]. HATs add acetyl groups to the lysine residues of histone tails, leading to decondensation of the chromatin and increased gene expression, while HDACs remove acetyl groups, leading to suppression of gene transcription [132]. Therefore, chromatin structure modifications caused by acetylation/deacetylation alter the expression of target genes, affect transcriptional activation, and regulate chromatin condensation and its availability [133,134,135]. Both HDACs and HATs play important roles in radiotherapy through mechanisms involving radiosensitization, DNA repair, and gene expression modifications [99]. HDACis have been shown to enhance the radiosensitivity of cancer cells [6,136]. Since HDACs and HATs are involved in the DNA repair process, blocking their activity may represent an effective mechanism to stop the repair of damage to cancer cells caused by radiotherapy, thereby improving treatment outcomes [107,135,137,138]. In the following section, we will summarize the roles of HATs and HDACs in cancer therapy, highlight ongoing clinical trials, and discuss future directions in the context of radiopharmacy.

#### 3.2.1. HDACis

A major aspect of the role of HDACs in cancer treatment is associated with HDACis. HDACis represent a promising class of anticancer drugs that modulate chromatin structure and gene expression by blocking histone deacetylation. They have been shown to exert radiosensitizing effects [6], for example, in sarcomas, where HDACis augment the cellular response to radiation [139]. Furthermore, scientific reports have demonstrated that HDAC activity can also be used as a biomarker in cancer therapies [100,101]. More than a twofold increase in HDACs activity was observed in patients with invasive grade III breast carcinomas, and radioresistant patients presented high HDACs and low HATs activity after irradiation. Therefore, patients with breast cancer tumors can be designated for HDAC inhibitor-based radio-sensitization treatment, which may result in much more effective treatment [102].

Vorinostat, an inhibitor of class I and II of HDACs, is a promising anticancer agent that inhibits the proliferation of various cancer cell types, including breast carcinoma [140], sarcomas [139,141,142], or lymphomas [143,144]. The addition of vorinostat to radiotherapy in osteosarcoma cell lines induces radiation-induced apoptosis, causes cell cycle arrest, and inhibits cell proliferation and clonogenic survival, with similar effects in RMS cell lines [142]. Moreover, vorinostat enhances radiation in osteosarcoma and RMS cells by inhibiting the expression of radiation-induced DNA repair proteins (*Rad51* and *Ku80*) [141]. Both mechanisms lead to significant radiosensitization caused by using vorinostat. The combination of vorinostat with heavy ion radiotherapy in osteosarcoma models resulted in a marked delay of tumor growth due to increased rate of apoptosis, elevated expression of *p53* and *p21* (proteins associated with cellular responses to stress like DNA damage), and inhibition of proliferation and angiogenesis, compared with heavy ion radiotherapy alone [145]. That shows the radiosensitization effects of vorinostat are also in heavy ion radiotherapy. One approach involves the use of nanoparticles carrying chemotherapeutic drugs. Dual-target nanoparticles using both a cisplatin prodrug—that is, a modified, inactive form, typically a platinum (IV) complex, designed to be converted into the active platinum (II) cisplatin within cancer cells, aiming to improve efficacy, reduce toxicity, or overcome drug resistance [146]—and a HDACi (vorinostat) increased DNA damage, impaired cancer cell repair ability, and promoted the effects of radiotherapy [147,148]. The radiosensitivity of cancer cells can be increased by increasing DNA damage. Interestingly, in three pancreatic cancer cell lines (Su.86.86, MIA Paca-2, T3M-4) treated with vorinostat or CUDC-101, a small molecule that simultaneously inhibits epidermal growth factor receptor (EGFR), human growth factor receptor 2 (HER2), and subsequently irradiated after 24 h, increased the radiation sensitivity of pancreatic tumor cell lines in a dose-dependent manner, reduced proliferation and clonogenic survival, and increased radiation-induced apoptosis (diminished full length Poly (ADP-ribose) polymerase [PARP-1]) [136]. These findings indicate that both compounds act as effective radiosensitizers in pancreatic cancer. HDACs inhibition was also observed in peripheral blood mononuclear cells (PBMC), tumor biopsies, and paired skin biopsies from patients with intermediate or high-risk of HNSCC, treated with CUDC-101 and HDACis in combination with external beam radiation (70 Gy to gross disease over 7 weeks) [149]. Radiotherapy can upregulate both activating and inhibitory receptors, e.g., ribonucleic acid export 1 (Rae-1) and UL16-binding protein (ULBP) [148]. Rae-1 is a cell surface glycoprotein, which is expressed on cells and functions as a ligand for the activating receptor natural killer group 2 member D (NKG2D) [150]. ULBP is a family of cell surface glycoproteins that act as ligands for the activating receptor NKG2D, which is expressed on immune cells like natural killer (NK) cells and T cells [151]. Both Rae-1 and ULBP were found to be upregulated by radiotherapy and HDACi therapy, leading to increased NK cell cytotoxicity against cancer cells [152,153].

Vorinostat shows strong preclinical potential as a radiosensitizer, consistently enhancing radiation-induced apoptosis, cell-cycle arrest, and suppression of key DNA repair proteins across multiple tumor models. Its ability to boost heavy-ion radiotherapy efficacy and increase immune-activating NKG2D ligands further broadens its therapeutic relevance. However, its clinical applicability remains limited by the predominance of cell-line and animal data, uncertain translation to measurable patient benefit, potential systemic toxicity from broad HDACs inhibition, and challenges in drug delivery that may require nanoparticle formulations to achieve optimal tumor penetration. These factors highlight the need for more rigorous clinical trials, better-defined dosing windows, and biomarker-driven patient selection before vorinostat can be reliably integrated into radiotherapy-based cancer treatment.

#### 3.2.2. HATs

HATs, such as CREB-binding protein (CBP) and E1A binding protein p300 (p300), induce histone H3 lysine 27 acetylation (H3K27ac) at target gene promoters and modulate its transcription [154]. CBP/p300 has been shown to be overexpressed in many cancer and drug-resistant cancer cells, e.g., lung, breast, and small-cell carcinoma [100,101,103,104,105,106]. CBP/p300 activates oncogene transcription and induces cancer cell proliferation, survival, tumorigenesis, metastasis, immune evasion, and drug-resistance, and was shown to be a poor prognostic linked with increased tumor recurrence [104,106,107]. Inhibition of CBP/p300 has been shown to reduce H3K27ac, downregulate oncogene transcription, induce cancer cell growth inhibition and cell death, activate immune response, overcome drug resistance, suppress tumor progression in vivo, and enhance the anticancer efficacy of radiotherapy [107]. Therefore, new inhibitors for CBP/p300 epigenetic enzymes are promising novel anticancer agents for clinical translation because they can enhance the effectiveness of other cancer treatments, such as radiotherapy.

##### KATs

One of the emerging approaches integrating epigenetic mechanisms into radiotherapy is radiotherapy-triggered proteolysis-targeting chimeras (PROTAC) prodrugs. In this strategy, PROTACs and PROTAC-oriented targeted protein degradation (TPD) strategies offer more precise and effective treatment modalities for a variety of diseases, but mainly for cancers. One of the main degraders used lysine acetyltransferase (KAT) as a target, where degraders create ternary complexes comprising ligands for targeted proteins and E3 ubiquitin ligase, along with a connecting linker, allowing targeted degradation via a ubiquitination-dependent method [155]. In multiple breast cancer cell lines, upregulation of lysine acetyltransferase 7 (KAT7) has been reported, and its expression was negatively correlated with the survival of breast cancer patients [110]. KAT7 is involved in the control of cell survival, DNA replication, and transcription, and KAT7 inactivation relates to global loss of H3K14ac and decreased expression of a broad range of genes [109,156]. *KAT7* overexpression enhances the phosphoinositide 3-kinase (PI3K)/protein kinase B (Akt) signaling pathway, which is frequently dysregulated in breast cancer, promoting tumor growth, survival, and metabolism [157], and also contributes to radioresistance [110]. Moreover, silencing of *KAT7* has been shown to suppress breast cancer radioresistance in vitro, highlighting its importance in radiation response [110]. Therefore, acetyltransferases such as KAT7 and CBP/p300 hold great potential for clinical application.

##### NAT10

Another example of using HATs in cancer radiopharmacy is the development of radiotracers for tumor imaging and theranostic applications. N-acetyltransferase 10 (NAT10), a HAT that plays a key role in regulating gene expression by adding acetyl groups to histone proteins [158], can also modify RNA acetylation, particularly in rRNA and messenger RNA [111]. NAT10 is overexpressed in many cancers, such as breast, liver, colorectal, lung, bladder, cervical, and oral cancers, as well as multiple myeloma. Its overexpression is associated with tumor progression, resistance to apoptosis, and increased cell migration and invasion [33,112,113,114,115,127]. Therefore, inhibitors of NAT10, such as [^11^C] remodelin, have been used to develop positron emission tomography (PET) radiotracers for tumor imaging and theranostic applications in both mouse models [159] and breast cancer patients [160]. These agents enable a more targeted approach to visualizing and treating specific cancers by exploiting the elevated NAT10 expression, making this enzyme a promising target for the development of PET radiotracers. Another HAT-associated radiosensitizer is garcinol, a HAT inhibitor that blocks the chromatin remodeling process involved in the non-homologous and joining DNA repair pathway, which cancer cells use to repair radiation-induced DNA damage [105,161]. Treatment with garcinol prevents cancer cells (A549 lung and HeLa cervical carcinoma cells) from repairing radiation-induced DSBs, leading to cell death or senescence and inhibition of proliferation [144]. That makes garcinol another promising agent used in cancer treatment that improves the radiation effect.

Histone modifications play a central role in shaping chromatin structure, regulating DNA repair, and determining cellular radiosensitivity, and radiation disrupts these marks through ROS generation, altered enzyme activity, and direct chromatin damage. While both histone methylation and acetylation pathways offer promising therapeutic entry points, through HDACis, HAT inhibitors, KDM5 blockers, or radiotherapy-activated PROTACs, the evidence remains largely preclinical, with variable responses across tumor types and concerns about off-target effects and normal-tissue toxicity. Critically, although targeting histone-modifying enzymes shows strong potential to enhance radiotherapy and enable novel radiotheranostic strategies, translation to the clinic will require clearer biomarker-guided patient selection, better understanding of long-term epigenomic impacts, and rigorous validation in human studies. Table 4 presents a compressed and comprehensive summary of all the mentioned inhibitions of epigenetic mechanisms, and their role in anticancer treatments and diagnostics related to radiation.

## 4. Summary

As demonstrated, epigenetic modifications are not only among the mechanisms used in anticancer therapies but can also radiosensitize cancer cells. Epigenetic mechanisms, including changes in the activity or expression of epigenetic enzymes, can also serve as diagnostic tools. Moreover, epigenetic heterogeneity is widespread among different individual patients, tumor lesions, and disease stages, leading to the various radiation treatment outcomes. These molecular characteristics provide potential targets for emerging epigenetic drugs (epi-drugs), which can be combined with radiotherapy to achieve better treatment results. Given the complexity and heterogeneity of tumors, epigenetic mechanisms that can distinguish different cell fractions or even different cell cycles may represent promising new targets for diagnostics and therapy, which can be widely used in the field of radiopharmacy. Another possible link between anticancer therapies, diagnostics, and epigenetic mechanisms lies in the dynamic nature of epigenetic changes, which may serve as early markers of cellular changes, without permanent changes in DNA code and gene expression. On the other hand, the complexity, multiplicity, and balance of epigenetic modifications can be another method of diagnostic approaches that integrate multiple molecular markers to precisely assess tumor progression and patient condition. There is no doubt that the combination of epigenetic modulation with radiation provides more effective therapeutic outcomes than radiation alone. Figure 1 presents a summary of the link between the initial effects of radiation, the resulting epigenetic alterations, and their impact on clinical outcomes.

Future research on epigenetic radiosensitizers such as vorinostat must address several translational hurdles, including limited clinical evidence, heterogeneous tumor epigenomes, and challenges in achieving effective yet safe modulation of chromatin regulators in patients. While preclinical studies highlight strong radiosensitizing potential through impaired DNA repair, apoptosis induction, and immune activation, translating these findings into durable clinical benefit requires optimized drug delivery, biomarker-guided patient selection, and strategies that minimize systemic toxicity. Emerging radiotheranostic approaches—such as PET-based tracers targeting epigenetic enzymes like NAT10 or radiation-activated PROTACs that degrade HATs or HDAC-associated proteins—offer a promising pathway to overcome these barriers by enabling precise imaging, selective targeting, and combination regimens that exploit radiation-induced vulnerabilities. Together, these advancing technologies may bridge the gap between mechanistic insight and clinical efficacy, ultimately enabling personalized radiotherapy that integrates real-time epigenetic modulation.

Future challenges and directions include the need for more comprehensive studies to fully elucidate the complex interplay between radiation and epigenetic regulation, to identify specific and effective epigenetic-based therapies, and to translate these findings into clinical practice for improved patient outcomes. They may not only improve diagnostic approaches but also represent promising targets for the development of novel anticancer therapies in the future. However, the focus on developing new, mechanism-based epigenetic reagents should remain a key priority for researchers.

## Figures and Tables

**Figure 1 cells-14-01885-f001:**
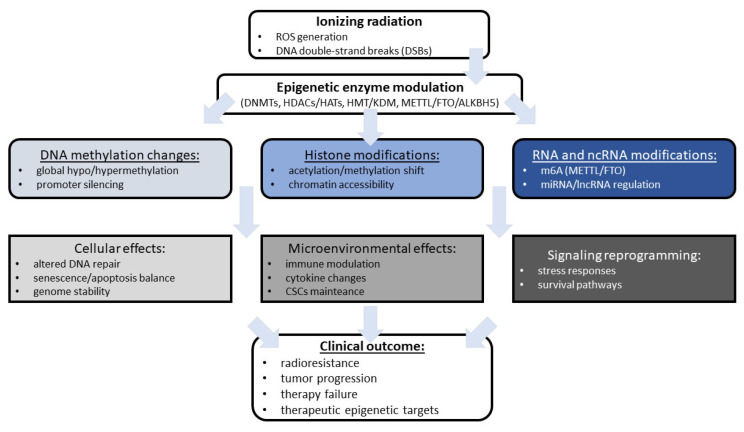
The summary of radiation-induced epigenetic remodeling and tumor response discussed in the review. ALKBH5, alkB homolog 5; CSCs, cancer stem-like cells; DNMTs, DNA methyltransferases; DSBs, double-strand breaks; FTO, fat mass and obesity-associated protein; HATs, histone acetyltransferases; HDACs, histone deacetylases; HMT, histone methyltransferases; KDM, lysine demethylase; m6A, N6-methyladenosine; METTL, methyltransferase-like; lncRNA, long non-coding RNA; miRNA, microRNA; ROS, reactive oxygen species.

**Table 1 cells-14-01885-t001:** The summary of epigenetic mechanisms involved in RNA modifications changed by cancer-related therapies and radiation.

Action Factor/Type of Enzyme	Specific Epigenetic Enzyme	Effect	Model/Cell Line	Citation
RNA methylases	METTL3	tumor progression, invasion, growth, drug resistance	in various cancers	[43]
		involved in UV-induced DNA damage response; elevated METTL3 recruitment	U2OS and HeLa	[44]
		elevated level	NSCLC	[45]
		METTL3 knockdown impairs the proliferation, migration, and invasion	NSCLC	[45]
	METTL14	enhance expression of miR-99a-5p, inhibited persistence	CSCs	[46]
		enhance expression of miR-99a-5p, reduces the radioresistance	ESCC	[46]
		suppressing *TRIB2* expression; facilitating the persistence of CSCs	ESCC	[46]
		METTL14 depletion enhances cell resistance to paclitaxel and doxorubicin; confers radioprotective effects	ALDH+ breast cancer stem cells	[48]
RNA demethylases	ALKBH5	downregulated expression *ALKBH5*, decreased survival in patients after irradiation; defects in DNA damage repair; decreased *CHK1* and *RAD51* expression	GBMSCs	[51]
	FTO	promotes resistance to chemoradiotherapy; upregulates β-catenin through mRNA demethylation; enhances ERCC1 activity	CSCC	[54]
		upregulated FTO	radioresistant NPC tissues and cells	[55]
		m6A modification of OTUB1; promotes the expression of *OTUB1*; inhibits radiation-induced ferroptosis	NPC cells	[55]

ALKBH5, alkB homolog 5; CHK1, checkpoint kinase 1; CSCs, cancer stem-like cells; CSCC, cervical squamous cell carcinoma; ERCC1, enhances excision repair cross-complementation group 1; ESCC, esophageal squamous cell carcinoma; FTO, fat mass and obesity-associated protein; GBMSCs, glioblastoma stem cells; HeLa, human cervical cancer cell line; m6A, N6-methyladenosine; METTL3, methyltransferase-like 3; METTL14, methyltransferase-like 14; NPC, nasopharyngeal carcinoma; NSCLC, non-small cell lung cancer; OTUB1, ubiquitin aldehyde binding 1; RAD51, RAD51 homolog 1; TRIB2, tribbles pseudokinase 2; U2OS, human osteosarcoma cell line.

**Table 2 cells-14-01885-t002:** The summary of epigenetic mechanisms involved in DNA modifications changed by cancer-related therapies and radiation.

Action Factor/Type of Enzyme	Specific Epigenetic Enzyme	Effect	Model/Cell Line	Citation
ROS		ROS decrease *ARID1A* expression through promoter methylation	ovarian cancers	[65,66]
ROS	DNMTs	ROS affect DNMTs and the expression of oncogenes and tumor suppressor genes	in various cancers, including lung, breast, colorectal, cervical, and prostate cancers	[25,91]
CpG island methylation		changes in CpG island methylation, caused by ADI-PEG20 (Arg1 inhibitor) and radiotherapy; complete tumor elimination by promoting macrophages/microglia recruitment and M1 repolarization	ASS1-negative and GBM-positive	[75]
genes methylation		82 differently methylated genes before and after radiotherapy; correlated degree of methylation with the radiation dose; enriched methylation in genes involved in immune response, proliferation and apoptosis pathways	in inoperable breast cancer patients	[81]
		changes in DNA methylation persist up to 7 months after exposure to radiation; DNA methylation alterations in heart-irradiated rats; differential expression of cardiac-relevant differentially methylated regions; higher expression for *SLMAP* correlated with hypomethylation; E2F6 inversely correlated with a decreased global longitudinal strain	in RICVD	[84,85]
mutations in IDH1/2		mutations in *IDH1/2* (observed in 80% of glioma) lead to the production of oncometabolite d-2-HG; aberrant in DNA methylation	HCT116 and U251	[88,89]
DNMTs	DNMT1	loss of DNA methylation; altered *DNMT1*, *MeCP2* and *MBD2* expression	in malignant MDA-MB-231	[69]
		low-dose X-ray irradiation increased global DNA hypomethylation; accumulation of DNA damage; reduced expression of *DNMTs*	in murine thymus	[67]
		affected cellular sensitivity to radiotherapy and DNMT levels	in malignant cancers	[68]
	DNMT3A/B	overexpression of *DNMT3A* and *DNMT3B* in primary tumor biopsies	in RMS	[70]
	DNMT3A	knockdown of DNMT3A followed by IR exposure increases radiosensitivity; activates the senescence program; upregulated p16 and p21	in embryonal RMS cells	[71]
	DNMT3B	knockdown of *DNMT3B* followed by IR exposure increases radiosensitivity; induces significant DNA damage; impairs the DNA repair mechanisms	in embryonal RMS cells	[71]
		*DNMT3B* silencing in response to radiation; DNA demethylation; enhanced the activation of cell cycle regulators p53 and p21; G1 phase cell cycle arrest; promoted apoptosis	NPC cells	[73]
		induced *DNMT3B* expression by X-ray radiation	PC3	[74]
DNMTs/CpG island methylation	DNMT3B	downregulation of *DNMT3B* by miR-145; DNMT3B knockdown increased expression of *miR-145* by CpG island promoter hypomethylation; sensitizing cancer cells to radiation	PC3	[74]

ADI-PEG20, Arginase 1 (Arg1) inhibitor; ARID1A, AT-rich interacting domain 1 A; ASS1, arginosuccinate synthetase 1; d-2-HG, d-2-hydroxyglutarate; DNMTs, DNA methyltransferases; DNMT1, DNA methyltransferase1; DNMT3A/B, DNA methyltransferase 3A/B; E2F6, E2F transcription factor 6; GBM, glioblastoma; HCT116, human colorectal carcinoma; IDH1/2, isocitrate dehydrogenase1/2; IR, ionizing radiation; MBD2, methyl-CpG binding domain protein 2; MDA-MB-231, triple-negative breast cancer (TNBC) cell line; MeCP2, methyl-CpG binding protein 2; NPC, nasopharyngeal carcinoma; p16, cyclin-dependent kinase inhibitor 2A; p21, cyclin-dependent kinase inhibitor 1A; p53, transformation-related protein 53; PC3, human prostate cancer cell line; RICVD, radiation-induced cardiovascular disease; RMS, rhabdomyosarcoma; ROS, reactive oxygen species; SLMAP, sarcolemma associated protein; U251, human glioblastoma multiforme.

**Table 3 cells-14-01885-t003:** The summary of epigenetic mechanisms involved in histone methylation and acetylation modulated by cancer-related therapies and radiation.

Modification	Action Factor/Type of Enzyme	Specific Epigenetic Enzyme	Effect	Model/Cell Line	Citation
histone modifications	ROS		ROS activate enzymes involved in histone control; modified expression of oncogenes and tumor suppressor genes	in various cancers	[25]
histone methylation	histone H4 methylation		low-dose X-ray irradiation decreased me3 of histone H4; reduced chromatin compactness	in the thymus and in vitro murine model	[67]
			decreased H4K20me3 and hyperacetylation of histone H4; lower expression of *Suv4-20h2*; increased malignant properties	MDA-MB-231 (breast cancer)	[69]
	H3K4-specific histone lysine demethylase	KDM5	overexpression of *KDM5* histone lysine demethylase	in different cancers, including breast, prostate, lung, or bladder carcinomas	[33,96,97]
			*KDM5B* knockdown promotes apoptosis; cell cycle arrest; autophagy; increases sensitivity to radiotherapy	KYSE-150 and TE-10 cells (ESCC)	[53]
			suppressed *KDM5B* expression; enhanced radioresistance; downregulation of *PTEN* expression	in NSCLC clinical samples	[98]
histone acetylation/deacetylation	HDACs/HATs		HDACs and HATs involved in radiosensitization, DNA repair, and gene expression modifications	in various models	[99,100,101]
			high HDAC and low HAT activity after irradiation	in radioresistant breast cancer patients	[102]
histone deacetylation	HDACs		doubled HDAC activity	in patients with invasive grade III breast carcinomas,	[102]
histone acetylation	HATs	CBP/p300	*CBP/p300* overexpressed	in many cancer and drug-resistant cancer cells lung, breast, and small-cell carcinoma	[103,104,105,106]
			CBP/p300 activate oncogene transcription and induce cancer cell proliferation; survival; tumorigenesis; metastasis; immune evasion; drug-resistance; poor prognostic and linked with increase tumor recurrence	in various cancers	[104,106,107]
		KAT7	control of cell survival; DNA replication and transcription	in colorectal cancer	[108,109]
			upregulation of KAT7, expression negatively correlated with survival of patients	in multiple breast cancer cell lines	[110]
			*KAT7* overexpression enhances the PI3K/AKT signaling pathway; promotes tumor growth, survival, metabolism; contributes to radioresistance	in breast cancers	[110]
			silencing of *KAT7* suppress radioresistance in radiation response	in breast cancers	[110]
		NAT10	modify RNA acetylation in rRNA and mRNA;	in HeLa and HEK293 cells	[111]
			overexpressed in many cancers; associated with tumor progression, resistance to apoptosis, increased cell migration, invasion	in breast, liver, colorectal, lung, bladder, cervical, oral cancers, multiple myeloma	[112,113,114,115]

Akt, protein kinase B; CBP, CREB-binding protein; ESCC, esophageal squamous cell carcinoma; H3K4, histone H3 lysine 4; H4K20me3, histone 4, lysine 20 trimethylation; HATs, histone acetyltransferases; HDACs, histone deacetylases; HEK293, human embryonic kidney cell line; HeLa, human cervical cancer cell line; KAT7, lysine acetyltransferase 7; KDM5, lysine demethylase 5; KYSE-150, human esophageal squamous cell carcinoma (ESCC); MDA-MB-231, triple-negative breast cancer (TNBC) cell line; mRNA, messenger RNA; NAT10, N-acetyltransferase 10; NSCLC, non-small cell lung cancer; p300, E1A binding protein p300; PI3K, phosphoinositide 3-kinase; PTEN, phosphatase and TENsin homolog; ROS, reactive oxygen species; rRNA, ribosomal RNA; Suv4-20h2, suppressor of variegation 4–20 homolog 2; TE-10, human esophageal squamous cell carcinoma cell line.

**Table 4 cells-14-01885-t004:** A summary of the effects of inhibition of epigenetic mechanisms in anticancer treatments and diagnostic related to radiation.

Modification	Blocked Enzyme	Inhibitor	Effect	Model/Cell Line	Citation
RNA methylation	FTO	FB23-2	enhanced the sensitivity to radiotherapy	in NPC cells	[77]
	FTO		genetic and pharmacological inhibition of FTO enhanced the efficiency of radiotherapy, increased DNA damage, reduced efficiency of homologous directed repair, and decreased formation of RAD51 foci, no radiation-induced oral mucositis exacerbation	in head and neck cancer	[57]
DNA methylation	DNMTs	psammaplin A, 5-aza-2′-deoxycytidine, and zebularine	in combination with radiation, induces radiosensitivity	A549 and U373MG cell lines	[6]
		psammaplin A	increases the sub-G1 fraction; prolonged γH2AX expression, a marker of radiation-induced DSBs	A549	[6]
		MA17, phthalimido-alkanamide derivative,	radiosensitized; decreased DNMTs activity; decreased *FANCA* expression; increased apoptosis, autophagy, and DSBs formation compared to radiotherapy alone	U87MG, U373MG, U138MG, T98G (glioblastoma cell lines) and not affected NHA	[83]
histone methylation	KDM	RS3195, RS5033, and KDOAM-25,	increased sensitivity of breast cancer cells to IR; enhanced radiation-induced damage	in MCF-7 breast cancer cells overexpressing *KDM5B*	[96]
	KDM5B	GSK467	enhanced the H3K4me3 methylation of *PIK3C3* promoter and induced the expression of PIK3C3	in ESCC patients, presenting overexpression of *KDM5B*	[97]
histone deacetylation	HDACs		HDACi augments the cellular response to radiation	sarcomas	[139]
			radiotherapy and HDACis upregulate *Rae-1* and *ULBP*, increasing NK cell cytotoxicity against cancer cells		[152,153]
		vorinostat	inhibits the proliferation of various cancer cell types	breast carcinoma	[140]
				sarcomas	[139,141,142]
				lymphomas	[143,144]
		vorinostat and radiotherapy	radiosensitization; apoptosis; cell cycle arrest; inhibited cell proliferation and clonogenic survival; inhibited *Rad51* and *Ku80* expression	in osteosarcoma and RMS cell lines	[142]
		vorinostat with heavy ion radiotherapy	delay of tumor growth; increase rate of apoptosis; elevate expression of *p53* and *p21*; inhibit proliferation and angiogenesis	in osteosarcoma models	[145]
		vorinostat combined with nanoparticles carrying chemotherapeutic drugs	increase DNA damage; impair repair ability; promote radiotherapy effects	EMT-6 cells (mouse mammary carcinoma)	[147,148]
histone acetylation	CBP/p300,		reduce H3K27ac; downregulate oncogene transcription; induce cancer cell growth inhibition and cell death; activate immune response; overcome drug resistance; suppress tumor progression; enhance the anticancer efficacy of radiotherapy		[107]
	KAT7	TSA	global loss of H3K14ac; decrease expression of a broad range of genes associated with procentriole formation	HCT116	[108,109,156]
	NAT10	[^11^C]remodelin	used to develop PET radiotracers for tumor imaging and theranostic applications	mouse models	[159]
				breast cancer patients	[160]
	HATs	garcinol	blocks chromatin remodeling process and DNA repair pathway; leads to cell death or senescence; senescence and inhibition of proliferation from repairing radiation-induced DSBs	in A549 (lung cancer) and HeLa (cervical carcinoma)	[105,161]

A549, human lung adenocarcinoma epithelial cell line; CBP, CREB-binding protein; DNMTs, DNA methyltransferases; DSBs, double-strand breaks; EMT-6, murine mammary adenocarcinoma line; ESCC, esophageal squamous cell carcinoma; FANCA, FA complementation group A; FB23-2, FTO inhibitor; FTO, fat mass and obesity-associated protein; GSK467, KDM5B inhibitor; HATs, histone acetyltransferases; HCT116, human colorectal carcinoma cell line; HDACs; histone deacetylases; HDACis, histone deacetylase inhibitors; HeLa, human cervical cancer cell line; IR, ionizing radiation; KAT7, lysine acetyltransferase 7; KDM, lysine demethylase; KDOAM-25, KDM5 inhibitor; Ku80, ATP-dependent DNA helicase II subunit 2; MA17, DNMTs inhibitor; MCF-7, human breast cancer cell line; NAT10, N-acetyltransferase 10; NHA, normal human astrocytes; NK, natural killer; NPC, nasopharyngeal carcinoma; p21, cyclin-dependent kinase inhibitor 1A; p300; E1A binding protein p300; p53, transformation-related protein 53; PET, positron emission tomography; PIK3C3, phosphatidylinositol 3-kinase catalytic subunit type 3; Rad51, DNA repair protein RAD51 homolog 1; Rae-1, ribonucleic acid export 1; RS3195, KDM5 inhibitor; RS5033, KDM5 inhibitor; T98G, KDM5 inhibitor; TSA, trichostatin A, KAT7 inhibitor; RMS, rhabdomyosarcoma; U138MG, human cell line derived from a grade IV glioblastoma; U373MG, human glioblastoma cell line derived from a malignant brain tumor; U87MG, a human malignant glioblastoma; ULBP, UL16-binding protein; γH2AX, gamma-phosphorylated H2AX.

## Data Availability

No new data were created or analyzed in this study.

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
