# Peer review of "A New Look at the Role of Radiation-Related Epigenetic Mechanisms in Diagnosis and Anticancer Therapies"

_cells, 2025, doi:10.3390/cells14231885_

Round 1

Reviewer 1 Report

Comments and Suggestions for Authors

This review explores how radiation-induced epigenetic changes contribute to therapy resistance and metastasis, and investigates their dual utility as prognostic biomarkers and targets for sensitizing tumors to treatment. However, several concerns require attention before the manuscript can be considered for publication.

Comments:

  1. The review would be strengthened by a comparative analysis of related epigenetic modifiers, such as the distinct consequences of DNMT3A versus DNMT3B inhibition, moving beyond a descriptive listing of results.
  2. The review would benefit from a synthesizing conceptual framework that graphically links the initiating effects of radiation (e.g., ROS, DSBs) to the resulting epigenetic alterations and their ultimate impact on therapy resistance and tumor progression.
  3. The discussion of HDAC inhibitors like Vorinostat, while comprehensive in listing mechanisms and models, lacks critical analysis of their clinical applicability and limitations.
  4. To improve readability, the authors could employ subtitles to structure the results section around the key mechanisms of radiation-induced epigenetic change and their consequent effects on therapeutic outcomes.
  5. To strengthen the manuscript, the conclusion could be enhanced by incorporating sections on future research perspectives, significant translational hurdles, and the promise of novel approaches such as radiotheranostics that target epigenetic enzymes.
  6. The manuscript contains several grammatical errors that necessitate professional language editing. Examples include:

Abstract: "...ionizing radiation induce..." should be "…ionizing radiation induces..."

Page 2: "Radiation directly damage DNA..." should be "Radiation directly damages DNA..."

Page 3: "...which reverse this modifications..." should be "these modifications".

Page 9: The phrase "The proper form of isocitrate dehydrogenase (IDH)" is unclear. For scientific precision, please replace it with the standard terminology: "The wild-type isocitrate dehydrogenase (IDH) enzyme..."

Author Response

Thank you very much for taking the time to review our manuscript. Please find our detailed responses below, along with the corresponding revisions and corrections marked in the re-submitted files (in red).

Point-by-point response to Comments and Suggestions for Authors

Comments 1: “The review would be strengthened by a comparative analysis of related epigenetic modifiers, such as the distinct consequences of DNMT3A versus DNMT3B inhibition, moving beyond a descriptive listing of results.”

Response 1: Thank you for pointing this out. We added the requested sentence to the main text (page6, lines 252-256):

Therefore we can conclude, that DNMT3A primarily regulates the senescence program in RMS, with its silencing increasing p16 and p21 expression and promoting cellular senescence, while DNMT3B is more directly linked to DNA damage responses and repair, and its silencing leads to impaired DNA repair and enhanced radiosensitivity across multiple cancer types.”

Comments 2: “The review would benefit from a synthesizing conceptual framework that graphically links the initiating effects of radiation (e.g., ROS, DSBs) to the resulting epigenetic alterations and their ultimate impact on therapy resistance and tumor progression.”

Response 2: We prepared a new graphical element, which has been included in the Conclusion section (page 18, lines 631-640).

Figure 1 presents a summary of the link between the initial effects of radiation, the resulting epigenetic alterations, and their impact on clinical outcomes.

"Figure 1. The summary of radiation-induced epigenetic remodeling and tumor response discussed in review. ALKBH5, alkB homolog 5; CSCs, cancer stem-like cells; DNMTs, DNA methyltransferases; DSBs, double-strand breaks; FTO, fat mass and obesity-associated protein; HATs, histone acetyltransferases; HDACs, histone deacetylases; HMT, histone methyltransferases; KDM, lysine demethylase; m6A, N6-methyladenosine; METTL, methyltransferase-like; lncRNA, long non-coding RNA; miRNA, microRNA; ROS, reactive oxygen species."

Comments 3: “The discussion of HDAC inhibitors like Vorinostat, while comprehensive in listing mechanisms and models, lacks critical analysis of their clinical applicability and limitations.”

Response 3: We removed the short summary from the main text:

That demonstrates that combined radiotherapy and HDACi are promising techniques for further analysis.

Instead, following the reviewer’s suggestion, we added a critical analysis of the clinical applicability and limitations of Vorinostat. (page 14, lines 517-527).

Vorinostat shows strong preclinical potential as a radiosensitizer, consistently enhancing radiation-induced apoptosis, cell-cycle arrest, and suppression of key DNA repair proteins across multiple tumor models. Its ability to boost heavy-ion radiotherapy efficacy and increase immune-activating NKG2D ligands further broadens its therapeutic relevance. However, its clinical applicability remains limited by the pre-dominance of cell-line and animal data, uncertain translation to measurable patient benefit, potential systemic toxicity from broad HDAC inhibition, and challenges in drug delivery that may require nanoparticle formulations to achieve optimal tumor penetration. These factors highlight the need for more rigorous clinical trials, bet-ter-defined dosing windows, and biomarker-driven patient selection before vorinostat can be reliably integrated into radiotherapy-based cancer treatment.

Comments 4: “To improve readability, the authors could employ subtitles to structure the results section around the key mechanisms of radiation-induced epigenetic change and their consequent effects on therapeutic outcomes.”

Response 4: We implemented the reviewer’s suggestion by dividing long paragraphs into shorter, more readable units:

2.1.1. RNMTs (METTL3 and METTL14) (page 3, line 126);

2.1.2. RNA demethylases (ALKBH5 and FTO) (page 4, line 157);

2.2.1. ROS and DNA methylation (page 5, line 206)

2.2.2. DNMT1 (page 6, line 218);

2.2.3. DNMT3A/B (page 6, line 231);

2.2.4. DNA and CpG island methylation (page 7, line 257);

2.2.5. DNMTi (page 7, line 281);

2.2.6. Radiation-driven epigenetic changes (page 7, line 303);

3.1.1. X-ray irradiation and histone methylation (page 12, line 412);

3.1.2. HDMs (page 12, line 424);

3.2.1. HDACi (page 13, line 466);

3.2.2. HATs (page 14, line 528);

3.2.2.1. KATs (page 15, line 542);

3.2.2.2. NAT10 (page 15, line 561);

Comments 5: “To strengthen the manuscript, the conclusion could be enhanced by incorporating sections on future research perspectives, significant translational hurdles, and the promise of novel approaches such as radiotheranostics that target epigenetic enzymes.”

Response 5: We also added the following statement to the Conclusion section (page 19, lines 642-655):

Future research on epigenetic radiosensitizers such as vorinostat must address several translational hurdles, including limited clinical evidence, heterogeneous tumor epigenomes, and challenges in achieving effective yet safe modulation of chromatin regulators in patients. While preclinical studies highlight strong radiosensitizing potential through impaired DNA repair, apoptosis induction, and immune activation, translating these findings into durable clinical benefit requires optimized drug delivery, biomarker-guided patient selection, and strategies that minimize systemic toxicity. Emerging radiotheranostic approaches—such as PET-based tracers targeting epigenetic enzymes like NAT10 or radiation-activated PROTACs that degrade HATs or HDAC-associated proteins—offer a promising pathway to overcome these barriers by enabling precise imaging, selective targeting, and combination regimens that exploit radiation-induced vulnerabilities. Together, these advancing technologies may bridge the gap between mechanistic insight and clinical efficacy, ultimately enabling personalized radiotherapy that integrates real-time epigenetic modulation.”

Comments 6: “The manuscript contains several grammatical errors that necessitate professional language editing. Examples include:

Abstract: "...ionizing radiation induce..." should be "…ionizing radiation induces..."

Page 2: "Radiation directly damage DNA..." should be "Radiation directly damages DNA..."

Page 3: "...which reverse this modifications..." should be "these modifications".

Page 9: The phrase "The proper form of isocitrate dehydrogenase (IDH)" is unclear. For scientific precision, please replace it with the standard terminology: "The wild-type isocitrate dehydrogenase (IDH) enzyme..."

Response 6: Thank you for this suggestion. The manuscript has been sent for additional linguistic editing and proofreading by a native speaker. On page 9, we implemented the suggested change (line 321).

Reviewer 2 Report

Comments and Suggestions for Authors

Manuscript ID: cells-3996926

Title: A new look at the role of radiation-related epigenetic mechanisms in diagnosis and anticancer therapies.

The authors reported a review draft highlighting the radiation-related epigenetic mechanism in anticancer therapies. The topic is important and shows the relevance of Clinical translation and the collection of recent literature reports. Even though the topic is important, there are lots of issues with the manuscript before publication. So I recommend a major revision before publication.

Specific comments

  1. The review is largely descriptive and encyclopedic. It reports many facts but does not critically analyze, compare contradictions, or summarize gaps.
  2. Despite a strong title, the manuscript does not clearly identify what is new about this review.
  3. Its recommended to add a paragraph in the introduction defining: What is missing in current literature, how this review fills that gap and the unique angle (e.g., combining radiopharmacy perspective + epigenetics).
  4. Tables are long and dense; some entries appear redundant, with inconsistent formatting.
  5. For example, Table 1 mixes RNA and DNA components without clear separation headings.
  6. It's recommended to split the tables. adding clinical status (pre-clinical / clinical / approved) and relevance ranking
  7. Several sections are overly long, decreasing readability. Add figures/infographics, and split subsections into shorter sections.
  8. While mostly understandable, there are recurrent minor English phrasing issues:
  9. Examples: "What is important" → "Importantly"

"Withs cancers" → typo; Subject-verb disagreements in many places. The manuscript should undergo professional proofreading to avoid an impression of improper language.

  1. Clarify if ROS effects differ between photon, proton, carbon-ion, and alpha therapy.
  2. Clarify whether epigenetic effects are transient vs heritable across cell division.
  3. Ensure citation numbering order is maintained consistently (some duplicated or sequential mismatches).
  4. Improve figure caption clarity.
  5. Needs structural tightening.
  6. Add abbreviations at first use inside the text, not only at the end
  7. Some key elements, such as Non-coding RNA roles under radiation; an Epigenetic drug developmental pipeline; FDA-approved drugs and ongoing trials; dose-fractionation effects; Tumor microenvironment & immune epigenetics, etc., are missing.
Comments on the Quality of English Language

Lots of typo errors and language issues. Need to be improved. The manuscript should undergo professional proofreading.

Author Response

Thank you for reviewing our manuscript. Below, we provide detailed responses along with the corresponding revisions and corrections, which are marked in the resubmitted files (in red).

Comments 1: “The review is largely descriptive and encyclopedic. It reports many facts but does not critically analyze, compare contradictions, or summarize gaps.”

Response 1: We thank the reviewer for this comment. In response, we added an additional paragraph to the manuscript:

DNA methylation is a dynamic but heritable epigenetic modification that can be reshaped by radiation and ROS, influencing gene expression, genome stability, and cellular radiosensitivity. Radiation alters the activity of DNMTs, modulates CpG is-land methylation, and drives gene-specific methylation changes that can either pro-mote tumor progression or enhance radiosensitivity, depending on the cellular context. These radiation-induced methylation patterns not only affect cancer treatment out-comes through DNMT1, DNMT3A/B, and DNMTi-related mechanisms but also con-tribute to long-term normal-tissue effects such as radiation-induced cardiovascular disease.” (page 8, lines 333-340)

Histone modifications play a central role in shaping chromatin structure, regulating DNA repair, and determining cellular radiosensitivity, and radiation disrupts these marks through ROS generation, altered enzyme activity, and direct chromatin damage. While both histone methylation and acetylation pathways offer promising therapeutic entry points, through HDACi, HAT inhibitors, KDM5 blockers, or radio-therapy-activated PROTACs, the evidence remains largely preclinical, with variable responses across tumor types and concerns about off-target effects and normal-tissue toxicity. Critically, although targeting histone-modifying enzymes shows strong potential to enhance radiotherapy and enable novel radiotheranostic strategies, translation to the clinic will require clearer biomarker-guided patient selection, better under-standing of long-term epigenomic impacts, and rigorous validation in human studies. Table 4 presents compressed and comprehensive summary of all mentioned inhibitions of epigenetic mechanisms, and their role in anticancer treatments and diagnostics related to radiation.” (page 16, lines 582-594)

Future research on epigenetic radiosensitizers such as vorinostat must address several translational hurdles, including limited clinical evidence, heterogeneous tumor epigenomes, and challenges in achieving effective yet safe modulation of chromatin regulators in patients. While preclinical studies highlight strong radiosensitizing potential through impaired DNA repair, apoptosis induction, and immune activation, translating these findings into durable clinical benefit requires optimized drug delivery, biomarker-guided patient selection, and strategies that minimize systemic toxicity. Emerging radiotheranostic approaches—such as PET-based tracers targeting epigenetic enzymes like NAT10 or radiation-activated PROTACs that degrade HATs or HDAC-associated proteins—offer a promising pathway to overcome these barriers by enabling precise imaging, selective targeting, and combination regimens that exploit radiation-induced vulnerabilities. Together, these advancing technologies may bridge the gap between mechanistic insight and clinical efficacy, ultimately enabling personalized radiotherapy that integrates real-time epigenetic modulation.” (page 19, lines 642-654)

Comments 2 and 3: “Despite a strong title, the manuscript does not clearly identify what is new about this review.”

“Its recommended to add a paragraph in the introduction defining: What is missing in current literature, how this review fills that gap and the unique angle (e.g., combining radiopharmacy perspective + epigenetics).”

Response 2 and 3: We prepared a new paragraph in the Introduction (page 2, lines 60-80) to better clarify the novelty and scope of the review:

Despite growing evidence that radiation induces profound epigenetic alterations affecting both therapeutic response and normal-tissue toxicity, current literature largely treats radiobiology, epigenetics, and radiopharmacy as separate domains, leaving a fragmented understanding of how these mechanisms intersect in clinical practice. Existing reviews tend to focus either on molecular epigenetic changes or on radiopharmaceutical advancements, but rarely integrate these perspectives to explain how radiation-related epigenetic remodeling can be harnessed for both improved di-agnosis and targeted anticancer therapy. This review synthesizes both fields to outline how radiation-triggered DNA/RNA methylation changes, histone remodeling, and enzyme dysregulation can serve as actionable biomarkers or therapeutic targets within nuclear medicine. It uniquely emphasizes the emerging potential of radiotheranostics targeting epigenetic enzymes and demonstrates how understanding radia-tion-epigenome interactions can guide the development of new diagnostic tracers and epigenetically informed radiosensitizing strategies. This interdisciplinary fram-ing—bridging epigenetic biology with radiopharmaceutical innovation—constitutes the main conceptual advance of the review. This work addresses that gap by providing a unified framework that links radiation-induced epigenetic mechanisms with their implications for radiotheranostics, radiosensitization strategies, and radia-tion-associated toxicities. By combining a radiopharmacy perspective with emerging epigenetic insights, we highlight new opportunities for precision imaging, personalized treatment planning, and development of epigenetically informed radiotherapeutic ap-proaches.”

Comments 4-6: “Tables are long and dense; some entries appear redundant, with inconsistent formatting.”

“For example, Table 1 mixes RNA and DNA components without clear separation headings.”

“It's recommended to split the tables. adding clinical status (pre-clinical / clinical / approved) and relevance ranking.”

Response 4-6: Thank you for this comment. We split the tables into a more reader-friendly layout, with a clearer division according to the epigenetic mechanisms discussed.

Table 1. The summary of epigenetic mechanisms involved in RNA modifications changed by cancer related therapies and radiation. (pages 4-5);

Table 2. The summary of epigenetic mechanisms involved in DNA modifications changed by cancer related therapies and radiation. (pages 8-10);

Table 3. The summary of epigenetic mechanisms involved in histone methylation and acetylation modulated by cancer related therapies and radiation. (pages 10-12)

Table 4. The summary of effect of inhibition of epigenetic mechanisms in anticancer treatments and diagnostic related to radiation. (pages 16-17)

We intentionally omitted clinical status, as mixing animal and human studies could introduce confusion and misleading interpretations. In addition, many cited works are based on in vitro models, including commercial and primary tumor-derived cell lines, which cannot be clearly assigned to a specific research stage.

Comments 7: “Several sections are overly long, decreasing readability. Add figures/infographics, and split subsections into shorter sections.”

Response 7: According to the reviewer’s suggestion, we divided paragraphs into shorter sections wherever possible:

2.1.1. RNMTs (METTL3 and METTL14) (page 3, line 126);

2.1.2. RNA demethylases (ALKBH5 and FTO) (page 4, line 157);

2.2.1. ROS and DNA methylation (page 5, line 206)

2.2.2. DNMT1 (page 6, line 218);

2.2.3. DNMT3A/B (page 6, line 231);

2.2.4. DNA and CpG island methylation (page 7, line 257);

2.2.5. DNMTi (page 7, line 281);

2.2.6. Radiation-driven epigenetic changes (page 7, line 303);

3.1.1. X-ray irradiation and histone methylation (page 12, line 412);

3.1.2. HDMs (page 12, line 424);

3.2.1. HDACi (page 13, line 466);

3.2.2. HATs (page 14, line 528);

3.2.2.1. KATs (page 15, line 542);

3.2.2.2. NAT10 (page 15, line 561);

Comments 8 and 9:  “While mostly understandable, there are recurrent minor English phrasing issues:

Examples: "What is important" → "Importantly"

"Withs cancers" → typo; Subject-verb disagreements in many places. The manuscript should undergo professional proofreading to avoid an impression of improper language.

Response 8 and 9: As recommended, the manuscript has been submitted for professional linguistic editing and proofreading by a native speaker.

Comments 10:  ” Clarify if ROS effects differ between photon, proton, carbon-ion, and alpha therapy.”

Response 10: In the manuscript, based on the available source materials, we focused specifically on photons (X-rays/γ-rays) and carbon-ion radiation. We showed that photon radiation generates immediate, diffusion-driven ROS bursts leading to widespread DNA oxidation, BER dysregulation, and epigenetic enzyme modulation, whereas carbon-ion radiation induces delayed, mitochondria-dependent ROS production in addition to high-LET DNA damage, resulting in prolonged oxidative signaling. Because the reviewed literature did not provide ROS-specific information for proton therapy or alpha-emitting radiopharmaceuticals, we did not expand on these modalities.

Comments 11:  “Clarify whether epigenetic effects are transient vs heritable across cell division.”

Response 11: In the review, we clarified that radiation-induced epigenetic changes may be both transient and heritable. We highlighted long-lasting DNA methylation changes observed in animal models (persisting up to 7 months after irradiation, Section 2.2.6) and in patients (detectable 6 months after radiotherapy, also Section 2.2.6). These modifications were presented as mitotically stable and functioning as a form of cellular “memory.” In contrast, histone modifications (e.g., temporary reductions in H4 trimethylation; Section 3.1.1) and ROS-mediated changes in enzyme activity (e.g., Section 3.2) appear to represent transient early responses that do not necessarily persist through multiple cell divisions. Moreover, as some authors did not provide information on the long-term reversibility of these modifications, we limited our analysis to results explicitly supported by the cited studies.

Comments 12-15:  “Ensure citation numbering order is maintained consistently (some duplicated or sequential mismatches).”

“Improve figure caption clarity.”

“Needs structural tightening.”

“Add abbreviations at first use inside the text, not only at the end”

Response 12-15: We addressed these issues critically by verifying citation accuracy, improving figure captions, refining the article structure, and reviewing and updating all abbreviations used in the main text. All added modifications and changes are highlighted in red in the main text.

Comments 16:  “Some key elements, such as Non-coding RNA roles under radiation; an Epigenetic drug developmental pipeline; FDA-approved drugs and ongoing trials; dose-fractionation effects; Tumor microenvironment & immune epigenetics, etc., are missing.”

Response 16: Given the broad scope of the topic, we chose to limit the manuscript’s focus. This review concentrates on the underexplored cross-talk between radiation and epigenetic mechanisms—specifically DNA and RNA methylation, as well as histone methylation and acetylation—from an epigenetic and molecular perspective. For this reason, some topics mentioned by the reviewer were not included. However, we fully agree that these topics are highly valuable and plan to address them in a separate manuscript in the near future.

Round 2

Reviewer 1 Report

Comments and Suggestions for Authors

The authors have addressed all of my comments and I recommend to accept the paper.

Reviewer 2 Report

Comments and Suggestions for Authors

The authors have revised the manuscript, and it can now be accepted for publication in Cells.

Comments on the Quality of English Language

The language can be improved.